# Resistance Loss in Cemented Paste Backfill Pipelines: Effect of Inlet Velocity, Particle Mass Concentration, and Particle Size

**DOI:** 10.3390/ma15093339

**Published:** 2022-05-06

**Authors:** Qiusong Chen, Hailong Zhou, Yunmin Wang, Xiaoshuang Li, Qinli Zhang, Yan Feng, Chongchong Qi

**Affiliations:** 1School of Resources and Safety Engineering, Central South University, Changsha 410083, China; qiusong.chen@csu.edu.cn (Q.C.); 205512082@csu.edu.cn (H.Z.); wangyunmin@csu.edu.cn (Y.W.); zhangqinlicn@126.com (Q.Z.); 2Sinosteel Maanshan General Institute of Mining Research Co., Ltd., Maanshan 243000, China; 3School of Civil Engineering, Shaoxing University, Shaoxing 312000, China; xsli2011@126.com; 4State Key Laboratory of Safety and Health for Metal Mines, Maanshan 243000, China; chongchong.qi@csu.edu.cn

**Keywords:** cemented paste backfill, pipeline transportation, resistance loss, CFD simulations

## Abstract

Cemented paste backfill (CPB), a technology placing the solid waste into mined-out stopes in the mine through pipeline transportation, has been widespread all over the world. The resistance loss is an important parameter for pipeline transport, which is significantly affected by the slurry characteristics. However, the coupling effect of inlet velocity (IV), particle mass concentration (PMC), and particle size (PS) has not been well evaluated and diagnosed. Hence, the CFD-based three-dimensional network simulation of CPB slurry flow in an L-shaped pipe at different combinations of the three parameters was developed using COMSOL Multiphysics software, and the findings were validated through a loop experiment. The results show that increasing IV and reducing PS will contribute to the homogeneity of the slurry in the pipeline, while the PMC presents little effect. The pipe resistance loss is positively correlated with IV and PMC and negatively correlated with PS. The sensitivity to the three parameters is IV > PS > PMC. In particular, the resistance loss is minimal at IV of 1.5 m/s, PMC of 72%, and PS of 1000 um. The calculation model of resistance loss regressed from simulation presented a high accuracy with an error of 8.1% compared with the test results. The findings would be important for the design of the CPB pipeline transportation and provide guidance in the selection of transfer slurry pumps, prepreparation of backfill slurry, and pipe blockage, which will improve the safety and economic level of a mine.

## 1. Introduction

The mining industry has long played a key role in economic development, employment, infrastructure, defense building, and the provision of basic raw materials to society [1]. At the same time, mining activities have caused many negative impacts on the environment and human health [2,3]. One of the gravest risks posed to the environment and public health is from the tailings and waste rock generated during the mining process [4,5]. One of the simplest and most cost-effective methods for mine waste disposal is cemented paste backfill (CPB) technology [6,7,8], which is widely used around the world [9,10,11,12]. Compared with conventional mining methods, CPB technology can improve the mining environment and reduce operational losses by effectively dealing with tailings contaminant stockpiles, controlling surface subsidence, and reducing the ore depletion rate [13,14,15,16].

CPB uses dewatered mine tailings and hydraulic binders mixed with water to produce a composite material that is transported to underground extraction areas or open pits [17,18,19]. Thence, it is obvious that pipeline transport is an important part of the CPB system [20,21], significantly affecting the stability of the backfill process. The resistance loss is one of the important indicators for evaluating the performance of pipeline transport. In practical engineering, the resistance loss is influenced by many factors, so it is necessary to determine reasonable technical parameters of pipeline transport to reduce the resistance loss and ensure safe, efficient, and stable mining production processes.

To date, several studies have been carried out on resistance loss during pipeline transportation. Kaushal et al. [22] used Ansys Fluent software to analyze the concentration distribution and pressure drop in pipeline flows with high concentrations of fine particles. Wang et al. [23] used Flow Science FLOW-3D software to study the self-flow conveying performance of paste filling using variations in the flow velocity distribution and pipe and elbow resistance loss as indicators. Wu et al. [24] used COMSOL Multiphysics software to study the combined effects of inner pipe diameter and flow velocity on the pressure drop in a fresh cement cinder–coal ash slurry pipe. Bandyopadhyay et al. [25] used Fluent to study the pressure drop in a small-diameter pipe. Duan [26] et al. used CFD and DEM software to investigate the effects of inlet slurry velocity and pipe inclination on the pressure drop and conveying capacity of the pipeline. The above researches have studied the factors affecting the resistance loss of slurries during pipeline transport, and most of them focused on one or two factors in a single analysis; the coupling influence of multiple factors has not been taken into account. To the best of our knowledge, there is a gap in existing literature regarding the variation of CPB resistance considering the couple effects of inlet velocity (IV), particle mass concentration (PMC), and particle size (PS).

Therefore, the computational fluid dynamics (CFD) simulation of CPB slurry was conducted using COMSOL Multiphysics software to explore the coupling effect of IV, PMC, and PS on resistance loss. The innovative work includes (1) analysis of the properties and mechanisms of IV, PMC, and PS on particle settling and resistance loss in more detail; (2) investigation of the coupling effect of IV, PMC, and PS on resistance loss; and (3) the establishment of the calculation model to calculate the resistance loss considering the IV, PMC, and PS, which was validated through a loop experiment.

## 2. Model Description

### 2.1. Mixture Model

The backfill slurry is a two-phase (solid–liquid) mixture [27]. In COMSOL, either a mixture model or an Eulerian model can be used to describe the phase characteristics of backfill slurry during flow in the pipe; however, in this study, the former was used, as it is simpler to implement and considerably less computationally intensive than the Eulerian model [28,29,30].
(1)$$\rho_{t}+\nabla \cdot \left(\rho u\right)=0$$
where $\rho_{t}$ is the density of phase t (units: kg/$m^{3}$), ρ is the average density of the mixture (units: kg/$m^{3}$), and u is the mass-averaged mixture velocity (units: m/s).

The momentum Equation for the mixture is
(2)$$\begin{array}{c} \rho j_{t}+\rho \left(j\cdot \nabla \right)j+\rho_{c}q\left(j_{\mathrm{slip}}\cdot \nabla \right)j=-\nabla p+\nabla \cdot \tau_{\mathrm{Gm}}+\rho g+F\\ -\nabla \cdot \left[\rho_{c}\left(1+\phi_{c}\varepsilon \right)u_{\mathrm{slip}}\;j_{\mathrm{slip}}^{T}\right]\\ -\rho_{c}\varepsilon \left[\left(j\cdot \nabla \right)j_{\mathrm{slip}}+\left(\nabla \cdot \left(D_{\mathrm{md}}\nabla \phi_{d}\right)\right)j+j_{\mathrm{slip}}\;m_{\mathrm{dc}}\left(\frac{1}{\rho_{c}}-\frac{1}{\rho_{d}}\right)\right] \end{array}$$
where $j_{t}$ is the average velocity of the volume mixture in phase t (units: m/s), j is the velocity vector (units: m/s), $j_{\mathrm{slip}}$ is the slip flux (units: m/s), *q* is the reduced density difference (dimensionless), p is the pressure (units: Pa), $\tau_{\mathrm{Gm}}$ is the sum of the viscous and turbulent stresses (units: kg/(m·s^2^)), *F* is any additional volume force (unit: N/m^3^), $u_{\mathrm{slip}}$ is the slip velocity vector between the two phases (units: m/s), $j_{\mathrm{slip}}^{T}$ is turbulent slip flux (units: m/s), $D_{\mathrm{md}}$ is the turbulent dispersion coefficient (units: m^2^/s), $m_{\mathrm{dc}}$ is the mass transfer rate from the dispersed to the continuous phase (units: kg/(m^3^·s)), $\phi_{c}$ is the volume fraction of the continuous phase, $\phi_{d}$ is the volume fraction of the dispersed phase, $\rho_{c}$ is the continuous phase density (units: kg/$m^{3}$), and $\rho_{d}$ is the dispersed phase density (units: kg/$m^{3}$).

The slip flux is defined for convenience in Equation (2) is
(3)$$j_{\mathrm{slip}}=\phi_{d}\phi_{c}u_{\mathrm{slip}}$$

The turbulent dispersion coefficient *D*_md_ in Equation (2) is
(4)$$D_{\mathrm{md}}=\frac{\mu_{T}}{\rho \sigma_{T}}$$
where $\sigma_{T}$ is the turbulent particle Schmidt number (dimensionless). The particle Schmidt number is usually suggested a value ranging from 0.35 to 0.7. The default value in this article is 0.35. $\mu_{T}$ is turbulent dynamic viscosity.

The sum of viscous and turbulent stresses, $\tau_{Gm}$ in Equation (2), can be written as:(5)$$\tau_{Gm}=\mu \left(\nabla j+{\left(\nabla j\right)}^{T}\right)-\frac{2}{3}\mu \left(\nabla \cdot j\right)I$$
where μ is the mixture viscosity (units: Pa·s) and I is the identity matrix.

The transport equation for the volume fraction of the dispersed phase:(6)$$\frac{\partial }{\partial t}\left(\phi_{d}\rho_{d}\right)+\nabla \cdot \left(\phi_{d}\rho_{d}u_{d}\right)=\nabla \cdot \left(\rho_{d}D_{\mathrm{md}}\nabla \phi_{d}\right)-m_{\mathrm{dc}}$$

### 2.2. The k-ε Turbulence Model

The k-ε turbulence model is chosen to describe the flow state of the backfill slurry in the pipe. The k-ε turbulence model is one of the most commonly used turbulence models for industrial applications, with strong convergence and adaptability to meshes, and contains two transport equations for turbulent kinetic energy k and turbulent dissipation rate ε [31,32].

The transport Equation for *k*:(7)$$\rho \frac{\partial k}{\partial t}+\rho u\cdot \nabla k=\nabla \cdot \left(\left(\mu +\frac{\mu_{T}}{\sigma_{k}}\right)\nabla k\right)+P_{k}-\rho \varepsilon$$
where the production term is
(8)$$P_{k}=\mu_{T}\left(\nabla u:\left(\nabla u+{\left(\nabla u\right)}^{T}\right)-\frac{2}{3}{\left(\nabla \cdot u\right)}^{2}\right)-\frac{2}{3}\rho k\nabla \cdot u$$

The transport Equation for ε is
(9)$$\rho \frac{\partial \varepsilon }{\partial t}+\rho u\cdot \nabla \varepsilon =\nabla \cdot \left(\left(\mu +\frac{\mu_{T}}{\sigma_{\varepsilon }}\right)\nabla \varepsilon \right)+C_{\varepsilon 1}\frac{\varepsilon }{k}P_{k}-C_{\varepsilon 2}\rho \frac{\varepsilon^{2}}{k}$$

The closure factors used in Equations (7) and (9) are shown in Table 1 [33].

### 2.3. Flow Domain and Boundary Conditions

Figure 1 illustrates the geometric flow domain of the pipe, in which the inlet pipe and outlet pipe are 5 m and 20 m long, respectively, connected by a 90° pipe bending with a radius of curvature of 0.5 m and 0.4 m near the exterior and interior of the pipe, respectively. The diameter of the pipe mouth is 0.1 m. In addition, the geometry meshed into a structured network. The mesh sizing was chosen according to the ‘based on fluid dynamics’ setting in COMSOL, and the total number of meshes was 1,033,274, with a maximum size defined as 2.70 × 10^−2^ m and a minimum size of 8.07 × 10^−3^ m. The average mesh mass was 0.81, indicating good meshing. The backfill slurry is subjected to gravity (g = 9.8 m/s^2^) in the negative z-direction. The boundary conditions for the mixture turbulence model were selected as the inlet velocity, outlet pressure (with a relative inlet value of 0), and the no-slip condition at the pipe wall.

### 2.4. Simulation Scenarios

For the different simulation runs conducted in this work, the IV of the backfill slurry was set at 1.5 m/s, 2.5 m/s, and 3.5 m/s, the PMC at 68%, 70%, 72%, and 74%, and the PS at 150 μm, 500 μm, 700 μm, and 1000 μm. These three variables are combined to form 64 simulation scenarios to investigate the change in resistance loss in the bend and outlet pipe sections.

### 2.5. Model Validation

The reliability of the simulation results was ensured by validating the model with the experimental data from the literature [34]. The transport characteristics of a slurry in a horizontal pipe with a diameter of 54.9 mm and a length of 3.3 m were simulated. In this scenario, the simulation velocities were determined to be 1 m/s, and the concentrations of spherical glass beads (PS = 125 μm) in the slurry were determined to be 10%, 20%, and 30%, respectively. Figure 2 shows a comparison between the results from the numerical simulations and the loop experiments. There is good agreement between the two sets of data, with an error of <8%, indicating that the numerical simulations are acceptable for use in pipe transportation research.

## 3. Results and Discussion

### 3.1. Settling of Particles under Different Factors

The knowledge of the particle concentration and velocity distribution in the pipeline under different conditions (IV, PMC, PS) and their variation helps to understand the resistance loss mechanism during backfill slurry transport in the pipeline [35]. The flow state of the backfill slurry in the straight pipeline becomes progressively more stable during long-distance pipeline transport, so the settlement of particles in the middle section of the outlet pipeline was chosen to be studied in this model. Figure 3 shows the concentration distribution of the backfill slurry in the middle section at different IVs at a PS of 150 um and an initial PMC of 68%. It is evident in Figure 3a that the PMC difference from the top to the bottom of the pipe decreases as the velocity increases from 1.5 m/s to 3.0 m/s, and the particles are more evenly distributed in the cross section. To better illustrate this situation, Figure 3b shows the PMC variation curve for the middle cutoff line on the cross section. The results show that the PMC increases with increasing IV at the same level for Heights > 0.575 m and decreases with increasing IV at the same level for Heights < 0.575 m. For example, the PMC at the top of the section (Height = 0.1 m) is 47.91%, and the PMC at the bottom of the section (Height = 0 m) is 75.62% at an IV of 1.5 m/s—a difference of 27.71%. The PMC at the top of the section (Height = 0.1 m) was 58.94%, and the PMC at the bottom of the section (Height = 0 m) was 73.23% at an IV of 3.0 m—a difference of 14.29%. As can be seen from Figure 3c, since the greater the IV at the same height, the greater the normal velocity (NV) of the particles, which causes the particles to continue moving in the direction of the fluid without easily detaching, so the smaller-size particles settle faster, making the PMC above the pipe decrease and the PMC below increase. In Figure 3d, the tangential velocity (TV) of the particles decreases with increasing IV at Heights > 0.063 m, resulting in faster settling of particles with small PS. The TV of particles increases with increasing IV at heights < 0.063 m, making particles with small PS settle slower. However, the NV of the particles is numerically much greater than the TV with more influence.

Figure 4 shows the concentration distribution of the backfill slurry in the middle section at different PMCs at a PS of 150 um and an initial PMC of 68%. It can be seen from Figure 4a that the PMC difference from the top to the bottom of the pipe does not greatly change as the initial PMC increases from 68 to 74%. The PMC is higher at the same section height, which is more evident from the PMC variation curve on the vertical cutoff of the section in Figure 4b. In terms of the normal and tangential velocities of the particles, Figure 4c shows that the NVs of the particles at different initial PMCs follow the same trend and do not vary significantly in value. As shown in Figure 4d, the TV of the particles at the same level becomes smaller as the initial PMC increases, with slower settling of the particles leading to a lower PMC. Therefore, it is possible to reduce energy consumption by using a smaller PMC slurry for pipeline transport while ensuring that the strength of the backfill slurry meets the requirements.

Figure 5 shows the concentration distribution of the backfill slurry in the middle section at different PSs at an initial PMC of 70% and an IV of 3.0 m/s. Figure 5a shows the difference in PMC from the top to the bottom of the pipe as the PS increases from 150 um to 1000 um. Figure 5b shows the PMC variation curve on the vertical truncation of the section, where the PMC decreases with increasing PS at the same level for Heights greater than 0.059 m and increases with increasing PS at the same level for Heights less than 0.059 m. At a PS of 150 um, the PMC differences in the cross section are small, and the particles are more uniformly distributed, while at a PS of 500 um or more, the settling of the particles becomes significantly stronger, and the PMC differences at the top and bottom of the particle distribution are large. Since the larger PS particles drop faster at the top of the section, the PMC content is reduced, and the particles accumulate at the bottom of the pipe, causing difficulty in movement, making the PMC at the top low and the PMC at the bottom high. Figure 5c,d illustrate exactly this situation. The results show that the NV of the particles increases with increasing PS for Heights greater than 0.048 m and decreases with increasing PS for Heights less than 0.048 m. At the same level, the TV of the particles increases with increasing PS. In addition, at a PS of 150 um, the TV of the particles varies little at different heights, and the settling performance is weak.

### 3.2. Effect of Inlet Velocity on Resistance Loss

The resistance loss in the pipe was calculated based on Bernoulli’s Equation [36]:(10)$$\rho g\left(y_{1}+\frac{p_{1}}{\rho g}+\frac{v_{1}^{2}}{2g}\right)=\rho g\left(y_{2}+\frac{p_{2}}{\rho g}+\frac{v_{2}^{2}}{2g}\right)+h_{f}$$
where ρ is the average density of the mixture, $y_{1}\;\mathrm{and}\;y_{2}$ are the heights at positions 1 and 2, respectively (units: m), with corresponding average pressures $p_{1\;}\;\mathrm{and}\;p_{2}$, and average speeds $v_{1}\;\mathrm{and}\;v_{2}$, respectively, and $h_{f}$ is the resistance loss from position 1 to 2 (units: Pa).

Figure 6a shows the variation of the resistance loss at the bend and outlet pipe for different IVs at a PMC of 68% and a PS of 150 um. Overall, the resistance loss increases with increasing IV in both the bend and outlet pipe sections. For example, compared with a return fill slurry with an IV of 1.5 m/s, the resistance loss of a return fill slurry of 3.0 m/s increases by a factor of 2.43 in the bend section and 2.39 in the straight section. Comparing the bend with the outlet pipe section, the resistance loss at the bend is higher than that of the outlet pipe at the same IV. For the pipeline model of this study (Figure 1), it is clear from Equation (10) that the resistance loss of the backfill slurry in pipeline transport is mainly determined by the pressure drop. Figure 6b shows the pressure distribution in the bend section at different IVs. As the IV increases, from the inside to the outside of the bend, the pressure drop increases, and the pressure increases numerically, resulting in an increase in the resistance loss of the backfill slurry during the transport of the pipe, which can be observed in the maximum and minimum values of the pipe pressure. In terms of pressure distribution, the red area on the outside of the bend increases significantly at an inlet velocity of 3 m/s compared with an IV of 1.5 m/s, indicating that a larger pressure is required for the backfill slurry to pass through the bend at higher IVs.

In addition, the rheological properties of the Bingham model, Power Law model, and Herschel–Bulkley model [37,38] can be used to describe highly concentrated slurries; of these, the Bingham model is as follows:(11)$$\tau =\tau_{0}+\eta \frac{du}{dy}$$
where *τ* is the pipe wall shear stress (or internal friction) (units: Pa), *τ_0_* is the initial shear stress (units: Pa); *η* is the viscosity coefficient (units: Pa·s), and *du/dy* is the shear rate (units: s^−1^). Figure 6c shows the relationship between forces in the pipe for Bingham fluids, and it can be seen that an increase in IV leads to an increase in internal friction between the fluid layers, resulting in more energy being consumed by the movement of the backfill slurry, which in turn causes a higher pressure drop [39,40,41]; at the same time, the friction between the fluid and the pipe wall is positively correlated with IV, which increases the resistance to the movement of the fluid along the pipe wall [42,43].

### 3.3. Effect of Particle Mass Concentration on Resistance Loss

Figure 7a shows the variation of resistance loss at the bend and outlet pipe for different PMCs at an IV of 2.5 m/s and a PS of 500 um. Overall, the resistance loss increases with increasing PMC in both the bend and outlet pipe sections, and the variation is linearly related. For example, compared with a return fill slurry with a PMC of 68%, the resistance loss of a 74% return fill slurry increases by 20.3% in the elbow section and 27.9% in the straight section. Furthermore, comparing the bend with the outlet pipe section, the resistance loss at the bend is higher than that of the outlet pipe at the same PMC. Figure 7b shows the pressure distribution in the bend section at different PMCs. Similarly, as the PMC increases, from the inside to the outside of the bend, the pressure drop increases, and the pressure increases numerically. However, the area of the red zone does not change much for different PMCs compared to the parameter of IV, indicating that there is less influence of the PMC on the pressure gradient distribution in the pipe. In addition, as shown in Figure 7c, the viscosity increases as the PMC increases. For the bend, the viscosity is higher near the outside of the bend, requiring a higher pressure to allow the backfill slurry to pass through the outside of the bend, which corresponds to the pressure distribution in Figure 7b. For the outlet pipe, in the 0–2 m section, the viscosity starts to increase gradually near the inside of the pipe and starts to decrease on the outside. In the 2–20 m section, the viscosity increases on both sides of the pipe and decreases in the middle of the pipe, with the viscosity distribution gradually tending to stabilize. The increase in PMC leads to a greater frequency of entanglement of molecular chains between particles to the extent that the viscosity increases [44]. It can be seen that an increase in viscosity also leads to an increase in internal friction between the fluid layers and between the pipe walls by Equation (11), so that a greater pressure difference is required to overcome the internal friction to ensure the normal transport of the backfill slurry, leading to an increase in resistance loss.

### 3.4. Effect of Particle Size on Resistance Loss

Figure 8a shows the variation of the resistance loss at the bend and exit pipe for different PSs at an IV of 3.0 m/s and a PMC of 70%. The results show that the resistance loss at the bend increases slightly and then gradually decreases with the increase of PS, first from 150 um to 500 um, the resistance loss increases from 1.77 MPa/km to 1.82 MPa/km—an increase of 2.8%—then, from 500 um to 1000 um, the resistance loss decreases from 1.82 MPa/km to 1.44 Mpa/km—a reduction of 20.9%. The resistance loss at the outlet pipe decreases as the PS increases, increasing from 150 um to 1000 um, with the resistance loss decreasing from 1.18 MPa/km to 0.74 MPa/km—a 37.3% decrease. Figure 8b shows the pressure gradient distribution over the corresponding bends. It can be seen that the pressure from the inside to the outside of the bend decreases numerically as the PS increases but does not particularly affect the pressure gradient distribution. As shown in Figure 8c, the backfill slurry flowing inside the pipe will cause energy loss through collision and friction between the particles, as well as between the particles and the pipe wall, increasing the resistance loss. The finer particles are more numerous, making for a higher collision rate and a larger contact area for the same PMC but are smaller in collision strength than the coarser particles. Generally speaking, the increase in the number of collisions results in greater energy consumption compared with the collision strength of the particles [45,46], so the finer the particles, the greater the resistance loss. For the bend at the PS for 150 um, the resistance loss of the return filler slurry is less than 500 um, probably because the bend at the PS for 500 um particle flow direction changes, and the number of collisions with the pipe wall increases but the number of particles and 150 um particles is not much different in the energy consumption caused by the collision between particles slightly larger, resulting in a slight increase in resistance loss. However, as PS continues to increase, this gap increases significantly as the number of particles decreases further, making the number of collisions much lower.

### 3.5. Sensitivity Analysis and Calculation Model of Resistance Loss

The sensitivity study of the influencing variables enables a better understanding of the resistance loss of the backfill slurry during transport in the pipeline. It is a simple comparison of the importance of the three factors IV, PMC, and PS on the resistance loss of the horizontal pipe (outlet pipe) by calculating the extreme difference. The 16 sample groups (as shown in Appendix A) were selected for analysis by orthogonal analysis from the 64 sets of results from the simulations [47,48] (as shown in Appendix A). Figure 9 shows that the range for IV, PMC, and PS are 0.82, 0.22, and 0.39, respectively, and the greater the range, the stronger the influence on the resistance loss, so the magnitude of the sensitivity of the three factors is IV > PS > PMC. In actual backfill slurry pipeline transport, if the resistance loss is too high, it will be more effective to reduce IV in the first place compared with PS in PMC.

In addition, A global optimization algorithm in the 1stOpt software package was used to fit an equation for the relationship between the IV, PMC, and PS of the backfill slurry during transportation and the resistance loss, *y*, along the transportation path, as follows: (12)$$y=\frac{1-0.3102x_{1}-0.0126x_{2}+8.9432x_{3}}{-1.0482+0.0989x_{1}+0.0195x_{2}-0.0002x_{3}}+x_{1}-\frac{1}{{\left(1+x_{2}\right)}^{-0.2324}}$$
where $x_{1}$ is the IV, $x_{2}$ is the PMC, and $x_{3}$ is the PS in μm.

Figure 10a shows the image of the fitted function, where the x, y, and z axes represent $x_{1}$, $x_{2}$, $x_{3}$, respectively. Figure 10b shows a comparison between the 64 sets of simulation results and the values obtained from Equation (12). The results show that the fitted correlation coefficient is 0.98, indicating a good fit. Hence, Equation (12) can be used to predict the pipeline transportation of the backfill slurry.

### 3.6. Experimental Validation

In order to verify the accuracy and practicability of the calculation model regressed from simulation results, the loop experiment was conducted. The loop experiment procedure has been introduced in detail in the literature [15], and the experimental model is shown in Figure 11a,b. The tailings were collected from a lithium pyroxene mine located in Sichuan Province, China, and the particle size distribution is shown in Table 2. The mean particle size of tailings is 386 μm, calculated by the Rosin–Rammler function [49]. The main parameters used in the loop experiment were the PMC of 70.2% and the IV of 1.62 m/s and 1.79 m/s. The pipe for the backfill slurry has a measured length of 66 m and a lift height of 5 m.

Figure 12 shows a comparison between the results from the numerical simulations and the loop experiments. There is good agreement between the two sets of data, with an error of <8.1%. The application of numerical modeling to pipeline transport studies in this study is acceptable given the uncertainty in modeling and the complexity of the experiments.

## 4. Conclusions

In this study, a CFD method with mixture turbulence k-ε mode was used to investigate the transport of CPB slurry in the pipeline, in particular, the variation of the concentration distribution in the pipeline and the resistance loss for different slurry IVs, PMCs, and PSs. The main conclusions which may be drawn from the results obtained herein are as follows:Regarding the concentration distribution of particles in the backfill slurry pipeline, the IV and PS have a strong influence, while the PMC has a weak influence. The IV is increased, and the PS is reduced to reduce the settling performance of the particles;The pipe resistance loss in the backfill slurry is positively correlated with IV and PMC and negatively correlated with PS. The sensitivity of the three parameters is IV > PS > PMC. In particular, the resistance loss is minimal at IV of 1.5 m/s, PMC of 72%, and PS of 1000 um;The feasibility of the model was verified, and a calculated model of the resistance loss of the backfilled slurry at the straight pipe was fitted to IV, PMC, and PS, with a fitted correlation coefficient of 0.98, and the calculated model was verified by loop pipe experiments.

In this study, the influence of factors such as pipe diameter, bend curvature, and temperature on the pipe transport resistance of filled slurries was not considered; however, a larger pipe diameter may consume more energy to transport the slurry, and a high bend curvature may result in a more difficult slurry, which, of course, needs to be verified by actual simulations or experiments. Therefore, in future work, the model will be extended to consider the above factors regarding the pipeline transport of slurry.

## Figures and Tables

**Figure 1 materials-15-03339-f001:**
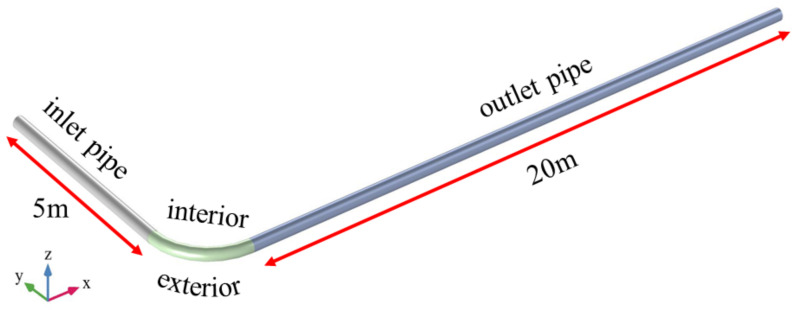
Geometric model of the pipe.

**Figure 2 materials-15-03339-f002:**
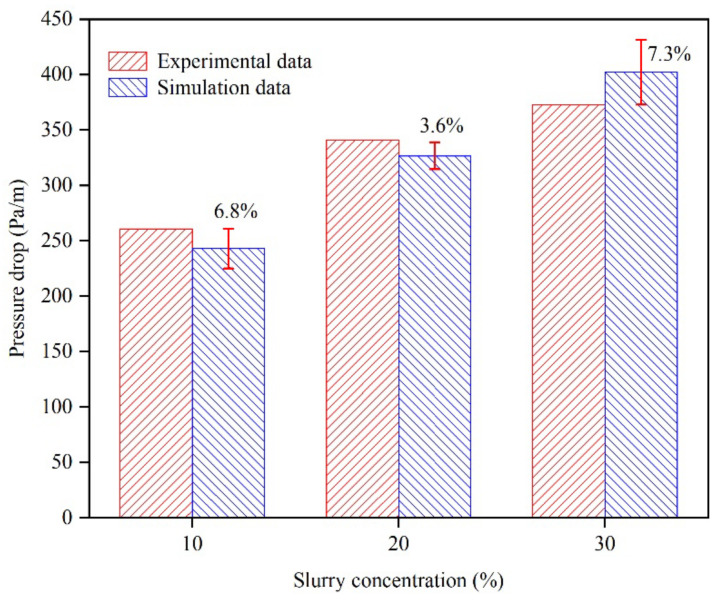
Validation of numerical simulation results with experimental data.

**Figure 3 materials-15-03339-f003:**
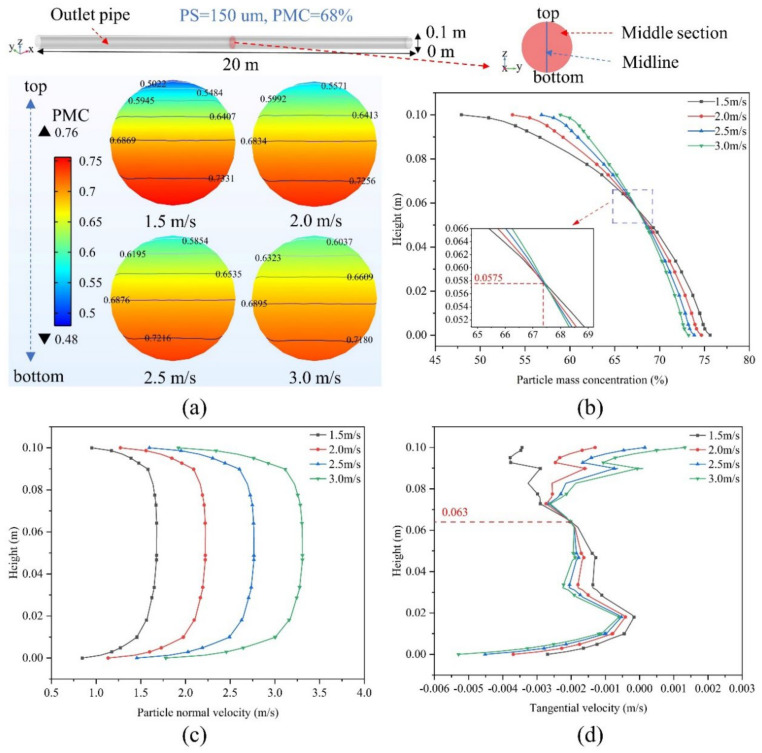
Particle concentration distribution of backfilled slurry in the middle section of the outlet pipe at different inlet velocities (example with PS = 150 um and PMC = 68%): (**a**) PMC distribution, (**b**) changes in PMC on the middle cutoff, (**c**) particle normal velocity (x-axis direction positive), (**d**) particle tangential velocity (z-axis direction positive).

**Figure 4 materials-15-03339-f004:**
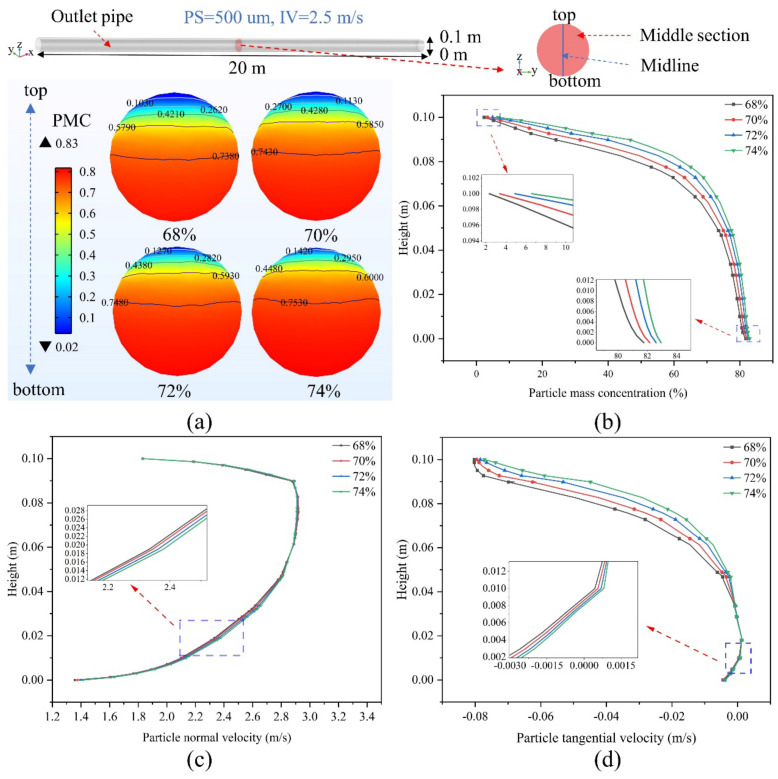
Particle concentration distribution of backfilled slurry in the middle section of the outlet pipe at different particle mass concentrations (example with PS = 500 um and IV = 2.5 m/s): (**a**) PMC distribution, (**b**) changes in PMC on the middle cutoff, (**c**) particle normal velocity (x-axis direction positive), (**d**) particle tangential velocity (z-axis direction positive).

**Figure 5 materials-15-03339-f005:**
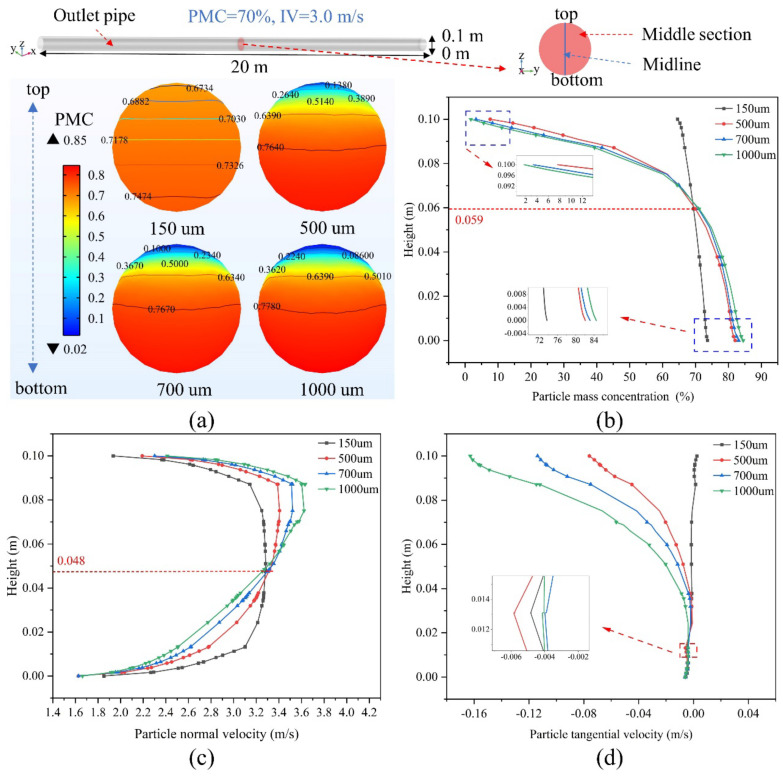
Particle concentration distribution of backfilled slurry in the middle section of the outlet pipe at different particle sizes (example with PMC = 70% and IV = 3.0 m/s): (**a**) PMC distribution, (**b**) changes in PMC on the middle cutoff, (**c**) particle normal velocity (x-axis direction positive), (**d**) particle tangential velocity (z-axis direction positive).

**Figure 6 materials-15-03339-f006:**
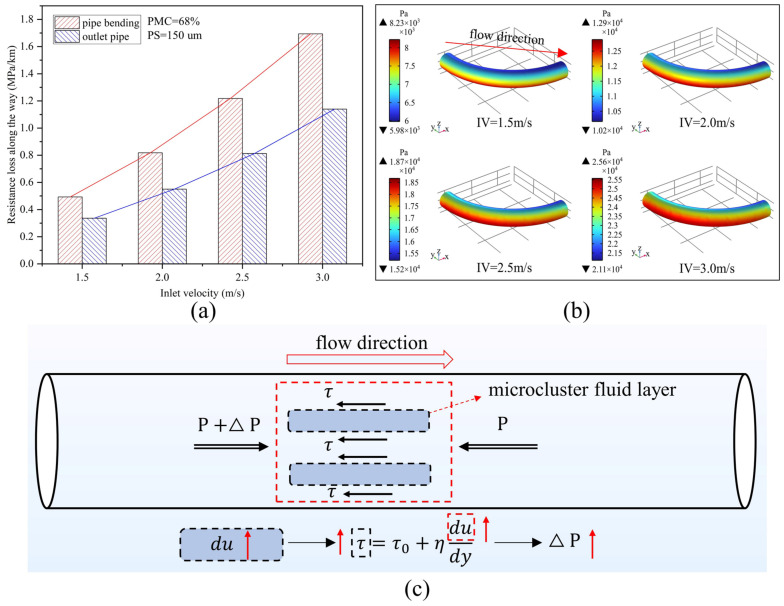
Pipeline transportation of backfill slurry with 68% particle mass concentration and 150 um particle size: (**a**) resistance loss at different inlet velocities; (**b**) pipe bending pressure distribution; (**c**) force analysis of the microcluster fluid layers.

**Figure 7 materials-15-03339-f007:**
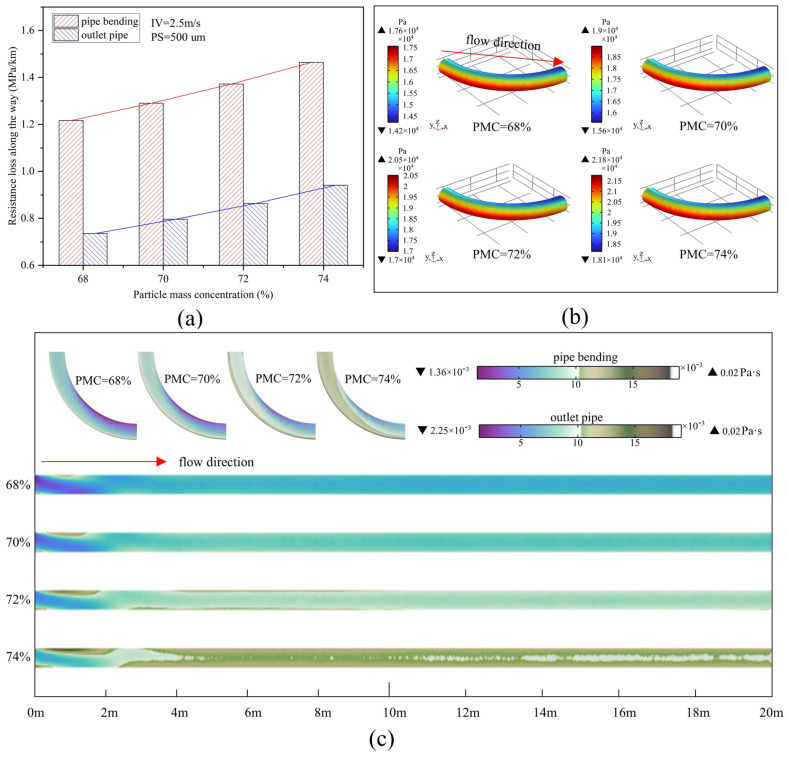
Pipeline transport of backfill slurry with 2.5 m/s inlet velocities and 500 um particle size: (**a**) resistance loss at different particle mass concentrations; (**b**) pipe-bending pressure distribution; (**c**) viscosity distribution in intermediate sections (bend and outlet pipe).

**Figure 8 materials-15-03339-f008:**
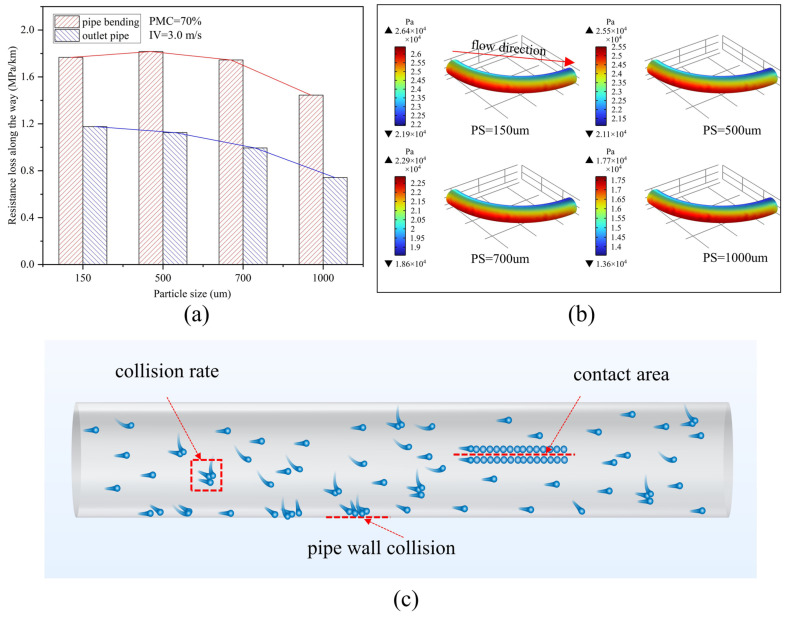
Pipeline transport of backfill slurry with 3.0 m/s inlet velocities and 70% particle mass concentration: (**a**) resistance loss at different particle mass concentrations; (**b**) pipe bending pressure distribution; (**c**) loss of movement of particles.

**Figure 9 materials-15-03339-f009:**
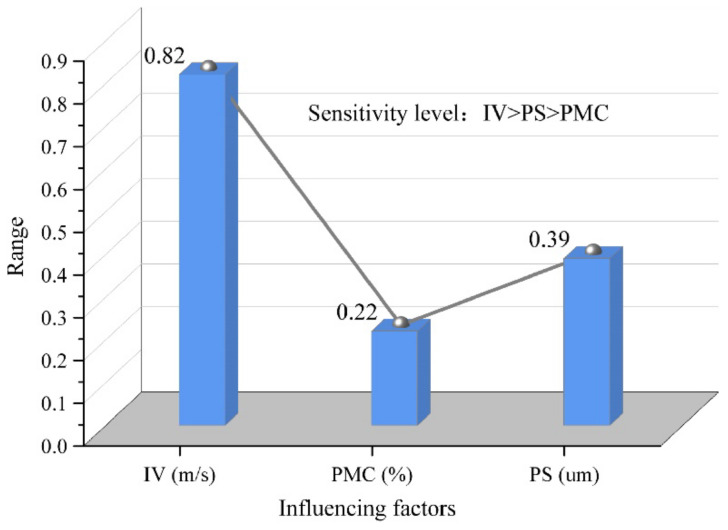
Sensitivity level.

**Figure 10 materials-15-03339-f010:**
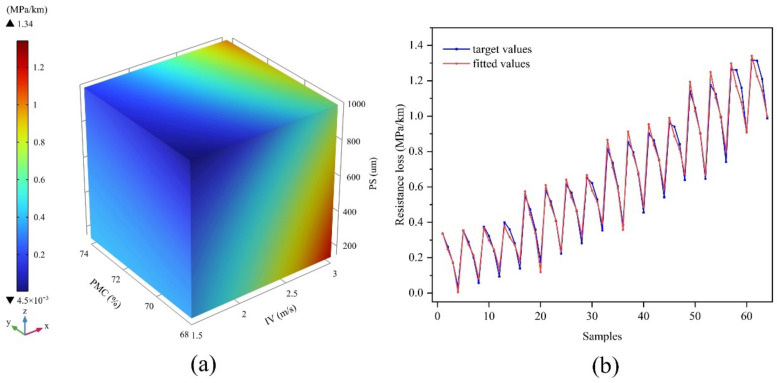
(**a**) The fitted function image and (**b**) comparison of target and fitted values.

**Figure 11 materials-15-03339-f011:**
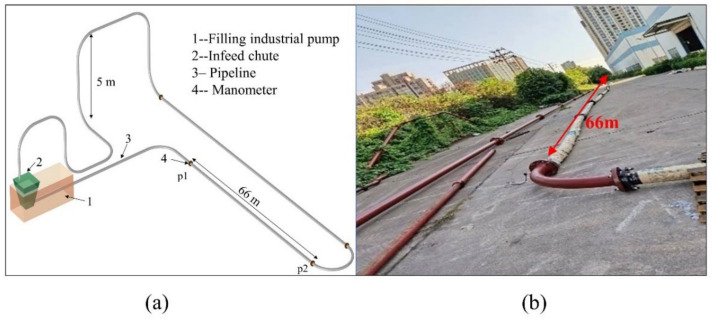
(**a**) Loop pipe experimental procedure and (**b**) loop pipe field experiment.

**Figure 12 materials-15-03339-f012:**
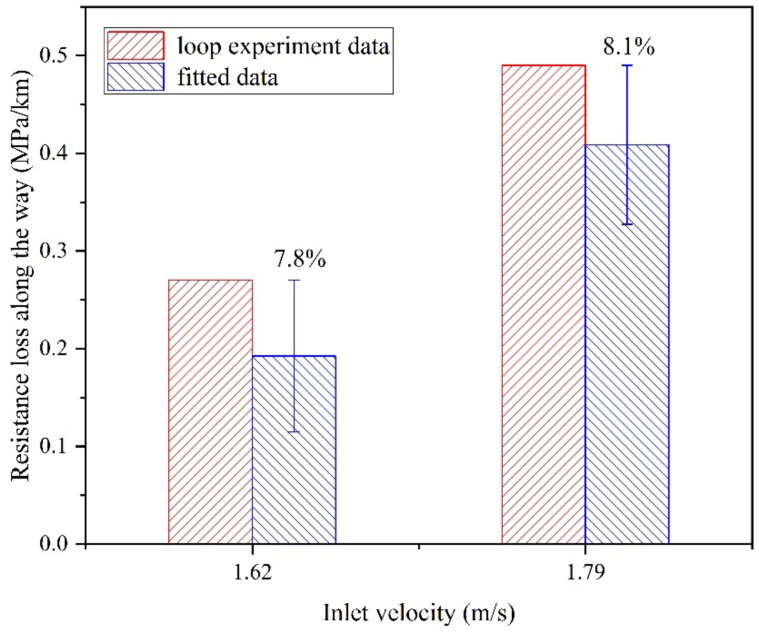
Comparisons between simulation and experiments.

**Table 1 materials-15-03339-t001:** The closure factors.

Constant	$\sigma_{k}$	$\sigma_{\varepsilon }$	$C_{\varepsilon 1}$	$C_{\varepsilon 2}$
**Value**	1.0	1.3	1.92	1.44

**Table 2 materials-15-03339-t002:** Test results for the particle size distribution of raw ore after crushing and sieving.

Particle Size/(mm)	+2.36	−2.36 ~ +1.18	−1.18 ~ +0.6	−0.6 ~ +0.3	−0.3 ~ +0.15	−0.15
Content/(%)	3.89	6.30	12.45	19.88	22.76	34.72

## Data Availability

The data presented in this study are available on request from the corresponding author.

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
