# Peer review of "Resistance Loss in Cemented Paste Backfill Pipelines: Effect of Inlet Velocity, Particle Mass Concentration, and Particle Size"

_materials, 2022, doi:10.3390/ma15093339_

Round 1

Reviewer 1 Report

The article contains interesting copmputer tests on transportation of cemented backfill which, both scientifically and technologically, forms the basis for a better understanding of the phenomena of resistance loss. Moreover, a very valuable part of the article is the experiment performed on the surface, the results of which were compared with the computer calculations. Below are some comments and suggestions:

  1. In the introduction it should be mentioned that the use of cemented paste backfill contributes to the reduction of operational losses by clean excavation of the deposit (https://doi.org/10.3390/en14227750);
  2. Lines 87 and 88, it should be checked numbers of equations.
  3. In table 1, it should be written the full names for ?k, ?ε, ??1, ??1, and write down the limit values ​​for these coefficients in the test.
  4. Line 114, it should be corrected the notation for the unit "... (g = 9.8 m / s2) ...", 2 superscript;
  5. Lines 19, 123, 148, 152, 156, 161, 163, 167, 171, 184, 189, 191, 192, 201, 202, 206, Figures 3-5, 217, 276, 278-282, 292-295 , 301, 360 it should be used a space between the digit and the unit.
  6. Figure 2, vertical axis, it should be corrected the notation for pressure drop (pa/m), it should be Pa/m.
  7. Figures 6a, 7a, 8a, 10a, 10b, 12 vertical axis, it should be corrected the notation for the unit (Mpa/km), it should be MPa/km.
  8. In Chapter 3.2 it should be written what other rheological formulas can be used in the calculations of the slurry flow, in particular with regard to the coefficients of viscosity.
  9. Table 2, the sign ~ should be replaced with another sign.
  10. In chapter 3.6, it should be written what was the maximum height of the cemented backfill lift in the experiment.
  11. In the summary, it should be written one conclusion relating to the identification of resistance loss in the filling flow.

Author Response

Response to Reviewer 1 Comments

Dear Reviewer,

Thanks very much for taking your time to review this manuscript. I really appreciate all your comments and suggestions! Please find my itemized responses below and my revisions/corrections in the re-submitted files.

Point 1: In the introduction it should be mentioned that the use of cemented paste backfill contributes to the reduction of operational losses by clean excavation of the deposit. (https://doi.org/10.3390/en14227750)

Response 1: Thank you for your nice suggestion. We have read the article and found it to be useful and helpful. We have added this in line 37 of the article and referenced the article [13].

Point 2:Lines 87 and 88, it should be checked numbers of equations.

Response 2: Thank you so much for your careful check. We have carefully checked the number of equations and found no errors.

Point 3:In table 1, it should be written the full names for σk, σε, Cε1, Cε2 and write down the limit values for these coefficients in the test.

Response 3: On this question, we have again carefully reviewed the relevant literature and the reference notes in the COMSOL software. The four variables σk, σε, Cε1, Cε2 are model constants without specific full names or restrictions. For this purpose, we have added a reference [33] in line 109.

(https://doi.org/10.1016/j.proeng.2015.01.224)

Point 4: Line 114, it should be corrected the notation for the unit "... (g = 9.8 m / s2) ...", 2 superscript.

Response 4: Thank you very much for your careful review. We have fixed this incorrect formatting in line 120.

Point 5:Lines 19, 123, 148, 152, 156, 161, 163, 167, 171, 184, 189, 191, 192, 201, 202, 206, Figures 3-5, 217, 276, 278-282, 292-295, 301, 360 it should be used a space between the digit and the unit.

Response 5: Thank you very much for your careful review. We have added spaces between the numbers and units in the corresponding rows as well as in the images.

Point 6:Figure 2, vertical axis, it should be corrected the notation for pressure drop (pa/m), it should be Pa/m.

Response 6: Thank you very much for your careful review. We have modified the format of the units in Figure 2.

Point 7:Figures 6a, 7a, 8a, 10a, 10b, 12 vertical axis, it should be corrected the notation for the unit (Mpa/km), it should be MPa/km.

Response 7: Thank you very much for your careful review. We have modified the format of the units in Figures 6a, 7a, 8a, 10a, 10b, 12 vertical axis.

Point 8:In Chapter 3.2 it should be written what other rheological formulas can be used in the calculations of the slurry flow, in particular with regard to the coefficients of viscosity.

Response 8: Thank you very much for your careful review. The Bingham model, the Power Law model, and the Herschel-Bulkley model have been used to describe the rheological properties of slurry flow, whose constitutive equation has the following scalar form.

The Bingham model:

The Herschel-Bulkley model:

The Power Law model:

where τ is the pipe wall shear stress (or internal friction) (units: Pa), τ0 is the initial shear stress (units: Pa); η is the viscosity coefficient, (units: Pa·s); and du/dy is the shear rate (units: s-1), n is the flow index or shear thinning exponent.

The above rheological equations can be used to describe fluid flow, and we have added the relevant notes to the manuscript and used the Bingham model as an example for analysis (lines 242 to 244).

Point 9:Table 2, the sign ~ should be replaced with another sign.

Response 9: Thank you very much for your careful review. We have replaced the symbols in the table with those in English format.

Point 10: In chapter 3.6, it should be written what was the maximum height of the cemented backfill lift in the experiment.

Response 10: Thank you for your nice suggestion. We have included this note in chapter 3.6 of the manuscript.

Point 11:In the summary, it should be written one conclusion relating to the identification of resistance loss in the filling flow.

Response 11: Thank you very much for your careful review. We have included an identifier for backfill slurry in the conclusion section.

Most sincerely,

Chongchong Qi (on behalf of all co-authors)

Reviewer 2 Report

The following comments must consider in revising the manuscript:

In the abstract it is mentioned that the findings from this work would be important for the design of the CPB pipeline transportation, which will improve the safety and economic level of a mine. Please discuss briefly how you measured the safety and economic level in this work. Also, please include a table of the data collected from the literature to validate the modeling. Furthermore, discuss briefly the effect different radius and curvature of the pipe in the modeling output.

Author Response

Dear Reviewer,

Thanks very much for taking your time to review this manuscript. I really appreciate all your comments and suggestions! Please find my itemized responses below and my revisions/corrections in the re-submitted files.

Most sincerely,

Chongchong Qi (on behalf of all co-authors)

Reviewer 3 Report

Dear Authors

I want to congratulate you on a wide research program with a perspective on recommendations for real-world applications. I value academic studies for their applicability. The subject is vital and important because many countries face the problem of mining and industrial (metallurgical, smelting) waste and its possible processing and reuse for a backfill. I appreciate any new developments in that field of knowledge, including the problem of cemented paste backfill transportation.

My general and detailed comments are given below with a set of suggested references. None of the proposed papers is co-authored by me so I have no personal gain from my reference suggestions. Please do not consider them as a mandatory list but rather just like a set of inspiring proposals.

  1. Your contribution is well composed and briefly follows the IMRaD structure of scientific contribution. Abstract covers strictly the content of the study.
  2. Introductory part is quite well written. It is comprehensive with regard to problem description in a global scale, but it is finally focused mainly on the issue under study.

I noticed that, in the introductory part, you refer to 20 references, including 13 Chinese, including 9 self-citations. It is absolutely clear to me that presented problem may be to some extent “local” and you have easier access to “local” references (including your former studies), but I have an impression that “State of the Art report” should cover wider experiences than only the Authors’. Please develop your research to provide more international background and experiences gathered worldwide for further analyses in the proceeding sections of your study. Usually, it wouldn't bother me, if the problem under study had just a local, Chinese importance. But your study is focused on issues of global importance (or, better to say, local problem appearing worldwide). That is why I'd suggest to widen your reference list, considering articles from other countries, based on other (maybe different) approach to the problem. That could be beneficial for the quality of your study, but also for its future citing potential, helping to focus other researchers’ attention on your work. I needed just a few minutes to find some (hopefully) relevant papers of European Authors (Polish, Russian, Slovak and Saudi Arabian) listed below.

  1. Concerning the entire section 2. I couldn’t find any important issues that I would question.
  2. Concerning results and discussion. Please try to discuss your results in the light of other researchers’ experience and ideas (if possible) referring again to “state of the art” presented in the introductory part.
  3. Concerning conclusions, please underline limitations of your study and perspectives for future developments. It helps the Readers to understand your position.
  4. Concerning the reference list, there is one major editorial issue: paper “RANS Modeling of Turbulent Flow and Heat Transfer of Non-Newtonian Viscoplastic Fluid in a Pipe” is referenced twice as [26] and [29]. Please correct.

As I mentioned in my comments to the introductory part, the vast majority of your references are Chinese and over 25% are self-citations. I do not question the relevance of your selection, however a wider geographic coverage would be appreciated. I just made a list of papers that attracted my attention and seem to be very relevant. Some refer directly to the problem of backfill transportation, other seem to be focused on possible use of mineral waste for production of backfill materials, others refer also to possible mineral extraction. Most of them are recent and open-access. Please feel free to make your own selection and/or your quick search in databases.

  1. Adigamov, A.; Rybak, J.; Golovin, K.; Kopylov, A. Mechanization of stowing mix transportation, increasing its efficiency and quality of the created mass. Transportation Research Procedia 2021, 57, 9-16. https://doi.org/10.1016/j.trpro.2021.09.019
  2. Adigamov, A.; Zotov, V.; Kovalev, R.; Kopylov, A. Calculation of transportation of the stowing composite based on the waste of water-soluble ores. Transportation Research Procedia 2021, 57, 17-23. https://doi.org/10.1016/j.trpro.2021.09.020
  3. Aleksakhin, A.; Sala, D.; Golovin, K.; Kovalev, R. Reducing energy costs for pipeline transportation. Transportation Research Procedia 2021, 57, 24-32. https://doi.org/10.1016/j.trpro.2021.09.021
  4. Kongar-Syuryun, C.; Tyulyaeva, Y.; Khairutdinov, A.M.; Kowalik, T. Industrial waste in concrete mixtures for construction of underground structures and minerals extraction. IOP Conf. Ser.: Mat. Sci. Eng. 2020, 869(3), 032004, https://doi.org/10.1088/1757-899X/869/3/032004
  5. Niemiec, D.; Duraj, M.; Cheng, X.; Marschalko, M.; Kubac, J. Selected black-coal mine waste dumps in the Ostrava Karvina region: An analysis of their potential use. IOP Conf. Ser.: Earth Environ. Sci. 2017, https://doi.org/10.1088/1755-1315/95/4/042061
  6. Rybak, J.; Adigamov, A.; Kongar-Syuryun, C.; Khayrutdinov, M.; Tyulyaeva, Y. Renewable-Resource Technologies in Mining and Metallurgical Enterprises Providing Environmental Safety. Minerals 2021, 11, 1145. https://doi.org/10.3390/min11101145
  7. Rybak, J.; Gorbatyuk, S.M.; Kongar-Syuryun, C.; Khayrutdinov, A.M.; Tyulyaeva, Y.; Makarov, P.S. Utilization of Mineral Waste: A Method for Expanding the Mineral Resource Base of a Mining and Smelting Company. Metallurgist 2021, 64, 851–861. https://doi.org/10.1007/s11015-021-01065-5
  8. Hefni, M.; Ahmed, H.A.M.; Omar, E.S.; Ali, M.A. The potential re-use of Saudi mine tailings in mine backfill: A path towards sustainable mining in Saudi Arabia. Sustainability 2021, 13, 6204. https://doi.org/10.3390/su13116204
  9. Zglinicki, K.; Szamałek, K.; Wołkowicz, S. Critical Minerals from Post-Processing Tailing. A Case Study from Bangka Island, Indonesia. Minerals 2021, 11, 352. https://doi.org/10.3390/min11040352
  10. Zglinicki, K.; Małek, R.; Szamałek, K.; Wołkowicz, S. Mining Waste as a Potential Additional Source of HREE and U for the European Green Deal: A Case Study of Bangka Island (Indonesia). Minerals 2022, 12, 44. https://doi.org/10.3390/min12010044

Best regards

Author Response

(The authors gave the same response as above.)

Round 2

Reviewer 3 Report

Respected Authors

The contribution was successfully improved according to my recommendations, As I mentioned in my first review, the entire research program is valuable and deserves publication.

Widening of the "State of the Art report" in the introductory part will certainly raise the citing potential of the study. 

I have no further comments concerning both: scientific and editorial merit of the study.

Sincerely yours